# Investigation of Asphalt Self-Healing Capability Using Microvasculars Containing Rejuvenator: Effects of Microvascular Content, Self-Healing Time and Temperature

**DOI:** 10.3390/ma16134746

**Published:** 2023-06-30

**Authors:** Qian Sun, Xin-Yu Wang, Sai Wang, Rong-Yue Shao, Jun-Feng Su

**Affiliations:** 1School of Information Engineering, Tianjin University of Commerce, Tianjin 300134, China; 2School of Mechanical Engineering, Tianjin University of Commerce, Tianjin 300134, China; wxy@tjcu.edu.cn; 3School of Material Science and Engineering, Tiangong University, Tianjin 300387, China

**Keywords:** microvascular, self-healing, asphalt, rejuvenator

## Abstract

The oily rejuvenator acted as the healing agent in microvasculars. A tensile test was designed to evaluate the self-healing efficiency of asphalt affected by microvascular number, self-healing time and temperature. It was found that the healing agent was slowly released through the microporous channels on the inner shell of the microvascular. The release modes of the agent can work together to improve the self-healing efficiency. The self-healing values of the three samples (asphalt, asphalt/microvasculars without rejuvenator and asphalt/microvasculars with rejuvenator) are 51%, 53%, and 71%. The self-healing capability of the asphalt samples with a healing agent is much greater than that of the other two without a healing agent at the same time. More microvascular rupture at the asphalt sample interface led to a higher self-healing efficiency. The self-healing efficiency values of the three samples (asphalt samples with one, two, and three microvasculars) are 52%, 67%, and 73%, respectively. The self-healing efficiency of the same sample increased during 1–3 days from 26% to 88% in one self-healing cycle. The self-healing efficiency value indicated that increasing the temperature improved each sample’s self-healing efficiency. The above trend of change also applies to the second self-healing process. A higher temperature reduces the resistance to molecular motion and accelerates the molecular action of bitumen and the healing agent. The time–temperature equivalence principle can be fully applied to comprehend asphalt self-healing.

## 1. Introduction

Asphalt is a common road construction material that is an organic cementitious material composed of various bituminous compounds. Apart from being brittle at low temperatures and sticky at high temperatures, it has self-healing properties [1]. However, under long-term ultraviolet, light, humidity, and other environmental conditions, the bituminous material’s self-healing function appears to be weakened. It requires external forces to improve its ability to heal itself. As a result, how to improve self-healing, delay the ageing of bituminous materials, and extend road service life has become a hot topic. It has been discovered that improving the self-healing function of asphalt material can reduce road maintenance costs and enable asphalt recycling [2]. This adheres to the green environmental protection principle and is critical to social and economic development.

Currently, there are three effective methods for self-healing bitumen. The first method is induction heating, which uses heat conduction to achieve a self-healing function. It incorporates steel wire into bituminous materials. It converts electrical energy from magnetic energy into thermal energy to heat bituminous materials [3]. The heat can put the bitumen molecules in a viscous flow state, and the molecular chains are entangled with each other to attain the purpose of self-healing. This method takes advantage of the temperature sensitivity of bituminous materials [4]. It has been reported that induction heating can prolong the life of marshal fatigue-damaged specimens by 31% [5]. Research has been carried out on the effect of mixed conductive additives on the distribution of steel fibres [6]. The bituminous material is heated unevenly, the heating temperature needs to be controlled easily, and this method accelerates the ageing of bituminous materials [7]. The second is the microencapsulation method, also known as in situ rejuvenation [8]. The rejuvenator is embedded in the microcapsules. When microcracks appear in the bituminous materials, the tip stress punctures the microcapsule shell, and the rejuvenator is released. It can be imagined that different molecular structures in asphalt materials lead to different molecular dynamics, and various molecular concentrations also affect molecular diffusion dynamics. The rejuvenator molecule is a small molecule structure; it gradually diffuses, penetrates and softens bitumen molecules [9]. Rejuvenators are generally engineered cationic emulsifiers, which consist of a lubricant and extender oils with a high content of maltene. It is reported that vegetable oil can also be applied as an asphalt rejuvenator agent [10,11]. At the same time, methods to control parameters such as microcapsule particle size distribution, shell thickness, and shell strength have also been proposed [12]. Yang et al. [13] studied the dynamic shear rheology and multi-stress creep properties of asphalt after adding microcapsules. Microcapsules’ small size means that the encapsulated rejuvenator amount is also small. It is tiny and meets the requirements of multiple repairs at the same site [14]. In addition, the microencapsulation method only works on the microcracks that are about to form and does not affect the cracks that have already begun to form. To avoid the shortcomings of the above two methods, the third method is the microvascular method in this paper.

It is reported in the literature that the microvascular self-healing system has been proven to recover large-scale damage to materials, especially in repairing the medium and multiple cracks of polymer materials [14]. Kato et al. [15] proposed a self-healing system using multiple short microvascular channels in the entire thickness direction of a composite. Due to the microvascular method’s strong practicability and simple operation, Li et al. [16] optimised and balanced the self-healing function and mechanical character of a matrix/microvascular material. Some scholars have even used light-sensitive self-healing microvascular epoxy resin composites for functional nanocontainers [17]. Interestingly, inspired by nature, the microvascular method is a biomimetic method of repair, just like that of the human blood system network. After the broken skin releases blood from the capillary, the platelets in the blood perform a coagulation function, and the wound gradually heals. Capillaries nourish the skin when it is not broken. Similarly, the rejuvenator is injected into the microvasculars to act as blood, and the microvasculars are laid on bituminous materials. After the microvasculars are broken by the tip stress of the microcrack, the rejuvenator is released. The bitumen molecules on both sides of the microcracks are reconnected under the moisturising behaviour of the rejuvenator to obtain the self-healing function. Additionally, microvasculars are prepared via a unique process, and the inner walls have uniform micropores. When the microvasculars are not damaged, the rejuvenator can also slowly permeate from the micropores to nourish the surrounding bitumen molecules. In this way, the purpose of delaying the ageing of bitumen molecules and reducing road maintenance costs is achieved.

In general, the microvascular self-healing bituminous materials method has been proven feasible. The researchers used raw materials and preparation methods to manufacture microvascular materials containing a rejuvenator. For example, Tabakovic [18] prepared Na-alginate-compartment microvasculars containing a rejuvenator via electrospinning to self-heal bituminous materials. Wu et al. [19] reported two different calcium alginate polymer microvessels. It was found that hollow calcium alginate polymer microvessels were more effective than compartment calcium alginate polymer microvessels were for self-healing asphalt. Shu et al. [20] synthesised a novel Ca-alginate/SiO_2_ self-healing microvascular encapsulation rejuvenator using a microfluidic control method to repair bituminous materials. Although SiO_2_ nanoparticles improve the stiffness of the Ca-alginate shell, they block the inner cavity and hinder the release of the rejuvenator due to the strong surface effect of the nanomaterials.

In previous work [21,22], our team successfully prepared microvasculars with polyvinylidene fluoride (PVDF) shells and filled them with a rejuvenator to heal bitumen. It was found that the self-healing efficiency value of pure bituminous material can reach 70% under a particular condition [21]. Subsequently, the survival rate and distribution of the PVDF microvasculars containing a rejuvenator in the asphalt mixture were explored under high-temperature processing [22]. The results showed that the microvasculars maintained good thermal and interfacial stability at a high temperature of 200 °C. Meanwhile, the team also observed that the microvasculars were uniformly distributed and kept intact in the asphalt mixture. The microstructure observation shows that the diffusion force of the rejuvenator in the asphalt mixture comes from the concentration gradient. When the microvascular is not broken, the rejuvenator molecule will not leak. Once the microvascular breaks, the rejuvenator molecule will quickly diffuse into the surrounding bituminous materials due to the effect of a concentration gradient. Due to the unique process of dry–wet spinning, microvessels are evenly distributed on the inner shell of the microvascular, which can meet the needs of long-term infiltration of the regenerant from the microvascular to nourish asphalt molecules [22]. Finally, the factors affecting the self-healing capability were studied, and an evaluation system of self-healing efficiency under certain conditions was established. According to the system, the self-healing efficiency of the microvasculars filled with an oily rejuvenator to the asphalt mixture can be up to more than 80%. Unfortunately, the self-healing mechanism needs to be better understood in terms of achieving a high value of self-healing efficiency, which motivates our group to continue to explore it.

Obviously, due to the complexity of the structure of microvascular materials, it is a systematic and complex research topic to use microvascular agents to achieve the self-healing function of asphalt. This work involves material preparation, structure characterisation, material performance research, mechanism exploration, and practical application. A preliminary inquiry was carried out in the early stage of this work. Firstly, microvascular was fabricated using PVDF material with a cavity structure. Then, the PVDF microvascular was filled with an oily rejuvenator, and both ends of the microvasculars were sealed with paraffin to avoid the outflow of the rejuvenator. The microvasculars were mixed into pure aged asphalt to study the self-healing capability. The experimental results show that with the help of the microvascular, the triggered microcracks disappeared rapidly, and the self-healing function of the aged asphalt was dramatically enhanced [22]. The above research results motivate a further exploration of the structure and morphology of microvasculars in asphalt mixtures. It was found that the healing molecules in the microvascular were not released in advance, even under the extrusion of aggregates. However, when external pressure is applied to the microvascular, the microvascular structure will be destroyed, and the oily healing agent will be released.

As we all know, practice promotes theory development, and perfect theory provides more powerful guidance for practice. Previous work shows that PVDF microvasculars containing a rejuvenator have a good self-healing effect on bituminous materials. The influence factors of high self-healing capability were analysed from an experimental and theoretical perspective to explore the underlying theory of self-healing capability. We combined empirical data with a self-healing mechanism to demonstrate the self-healing function of bitumen using microvasculars containing rejuvenators. This research will also discuss the influence of material structure on self-healing capability.

## 2. Materials and Methods

### 2.1. Materials

Sinopec supplied a bitumen sample with a 80/100 penetration grade. N, N- polyvinylidene fluoride (PVDF, 6010#; purity of >99.5%; density of 1.75 g/cm^3^) was provided by Tianjin Sinogo Co., Ltd., Tianjin Chemical Company, Tianjing, China. The oily rejuvenator (XT-1) was supplied by Tuoxin Chemical Company (Changzhou, China). Dimethylacetamide and others are chemically pure substances.

### 2.2. Preparation of Microvascular Containing Rejuvenator

A dry–wet spinning method was used to prepare microvasculars filled with oily healing agents [21]. PVDF (100 g) powders were dried in an electric oven at 50 °C for 2 h. Then, DMAc (400 g) and PVDF were mixed and heated in a sealed three-neck round-bottom flask to 50 °C for 4 h forming a homogeneous pouring solution. The casting solution remained under vacuum pressure for 24 h to eliminate bubbles. Then, the casting solution was poured into the spinning tank at a constant temperature of 60 °C, and the spinneret started to work under the pressure of N_2_ with A 0.2 MPa pressure. At the same time, the oil rejuvenator returned, driven by 0.2 MPa pressure flows into the core chamber. The casting fluid and core liquid were injected into the spinneret to form a microvascular structure filled with a healing agent. The spinning frame was passed through a coagulation bath kept at a temperature of 25 °C. The formed microvasculars were wound at room temperature on the guided wheel’s winding wheel. The ends of the microvasculars were sealed with a heat sealer. Finally, the microvasculars were immersed in distilled water at room temperature for future use.

### 2.3. Mixture of Microvascular in Bitumen

The long microvasculars were cut into 2.0 cm short ones. Both ends of the short microvasculars were sealed using paraffin to prevent the liquid healing agent from flowing out. A certain number of short microvasculars and aggregates (90%) were randomly mixed in bitumen (10%) and placed in a rectangular silica gel mould. The asphalt samples remained at −15 °C in the refrigerator for 48 h before application.

### 2.4. Observation of Oily Rejuvenator Release and Diffusion

The 2.0 cm microvasculars were injected with phosphor powder and irradiated under an ultraviolet lamp (ZF-1, LICHEN, Shanghai, China). The microvasculars, aggregates, and bitumen were mixed into silicone moulds to make the samples and refrigerated at 0 °C for 24 h. Fourier transform infrared spectroscopy (FT-IR, Perkin Elmer Spectrometer 100, Shanghai, China) was performed to measure the rejuvenator in the microvasculars and the pure rejuvenator, with a resolution of 4 cm^−1^ and a spectral record of a 250 cm^−1^–4000 cm^−1^ wave number in the range. In addition, the release behaviour of the rejuvenator was analysed using a fluorescence microscope (WSF400, Guangzhou Micro-optical Instrument Co., Ltd., Guangzhou, China) [8]. In addition, glass slide samples of the microvascular/bitumen composite were prepared, and the pieces were placed in an environment of 0 °C for 24 h. Liquid N_2_ was applied to trigger microcracks. The diffusion behaviour of the rejuvenator was also observed using fluorescence microscopy.

### 2.5. Design of Test Method for Self-Healing Capability

It was found that the tensile fracture test is one of the effective methods to measure the self-healing efficiency of materials. The self-healing efficiency is generally calculated according to the repeated fracture tensile values of microcracks generated at different healing times during the crack-healing process [22]. Microcrack is one of the leading causes of asphalt pavement damage. Understanding the behaviour of crack healing is crucial to predicting the service life of asphalt pavements. The experiment on fracture healing and re-fracture helped us to study the self-healing efficiency of this process. A method of repeated stretching is designed in this work. Based on this method, the self-healing feature of asphalt samples was investigated by comparing the self-healing efficiency under different temperatures and retention times.

Previous work [21] has reported that both the orientation and content of microvasculars in asphalt influence self-healing behaviour. It was found that the self-healing efficiency decreased slightly when the angle between the stretching direction and the microvascular was 15°–45° during the initial 1–2 self-healing cycles [21]. Meanwhile, increasing temperature and prolonging time naturally can improve the self-healing efficiency of asphalt, which obeys the rule of the time–temperature superposition principle. Self-healing behaviour is considered a process of crack closure and strength increase at a specific temperature for bitumen because of its viscoelastic character. To provide a quick and easy method for self-healing evaluation, the tensile test in this work is simplified into the following four rules:(1)The included angle between the microvascular arrangement direction and the tension direction of asphalt is entirely parallel.(2)The number of microvasculars at the asphalt test sample’s fracture interface (1 cm^2^) is 1–3.(3)The test temperature is room temperature.(4)Each asphalt sample is subject to two cycles, and each self-healing cycle is fixed at 24 h.

All tensile testing experiments in this paper were conducted under the same four conditions described above. Preliminary tests on microvascular self-healing asphalt mixtures’ mechanical properties laid the foundation for establishing later mathematical models. The self-healing efficiency (SHC) was measured through a designed tensile test using a tensile machine with a precision mechanical sensor. The model’s geometry ensures the stretch’s direction in the sample’s centre. Microvasculars, aggregates, and bitumen were mixed evenly at 80 °C and poured into a copper mould. After the above mixture was cooled to room temperature, it was stored in a refrigerator at −5 °C for 24 h. After de-moulding, the frozen sample was refrigerated and heated to room temperature. The tensile test was carried out at room temperature, and the tensile speed was uniform.

The sensor recorded the initial tensile strength at the break as (*T_b_*_0_). After the tensile fracture of the sample, the two parts of the fracture were spliced together according to the shape of the fracture surface. Then, it was placed in a 0 °C environment for 48 h. Another tensile test was repeated, and the tensile strength was recorded as (*T_b_*_1_). After the second self-healing of the sample in the same environment and simultaneously, the tensile strength was tested again as (*T_b_*_2_). The SHC value is calculated based on the ratio of the two tensile strength values as shown in Equations (1) and (2).
(1)$${\mathrm{SHC}}_{1}=\frac{T_{b1}}{T_{b0}}\times 100\%$$
(2)$${\mathrm{SHC}}_{2}=\frac{T_{b2}}{T_{b0}}\times 100\%$$
where SHC_1_ is the SHC value of the first self-healing cycle, SHC_2_ is the SHC value of the second cycle, *T_b_*_0_, *T_b_*_1,_ and *T_b_*_2_ are the original tensile breaking strength, the tensile strength after the first self-healing cycle, and the tensile strength after the second self-healing cycle.

## 3. Results

### 3.1. Self-Healing Mechanism of Microvascular/Asphalt Composites

The microvascular/asphalt mixture has various components, and its microstructure is more complex. There are many factors affecting the mechanism of microvascular asphalt. Due to the material structure’s complexity and the external environment’s variability (temperature, moisture and ultraviolet rays, etc.), many factors affect the self-healing effect of microvascular asphalt. Figure 1 illustrates the self-healing mechanism of asphalt using microvasculars. In the initial state, microvasculars and aggregates are mixed into the bitumen matrix (Figure 1a). In a typical area composed of microvasculars, aggregates, and bitumen, the microvasculars are basically in a stretched equilibrium state. There is no entanglement and cross-connection between a single microvascular and several microvasculars. A liquid rejuvenator acts as the self-healing agent, which is well-protected by the shell of microvasculars. After enduring harsh external conditions for a long time, the asphalt loses part of its properties and ages. Internal crazes or microcracks triggered by stress gradually spread, as shown in Figure 1b. The continuous expansion of microcracks leads to microvascular breakage. The intensity at the tip stress of the microcrack determines the number of perforated microcracks and the fracture speed of microcracks. Under the action of the capillary, the oily agent is released from the damaged microcracks. Driven by the concentration difference, the oily healing agent can rapidly penetrate the asphalt material around the microcrack (Figure 1c). The rejuvenator gradually diffuses around the bitumen. The bituminous material surrounding the microcrack is then infiltrated and softened by the wetting action of the self-healing agent. The viscosity of the bitumen dramatically decreases. As a result, the micro-crack is gradually healed (Figure 1d) under an appropriate temperature condition within a sufficient time.

The release rate and diffusion rate of an oily healing agent determine the whole rate of self-healing action. In contrast, the movement rate of the healing agent is determined by the driving force caused by the capillarity of microcracks and concentration gradient difference [22]. The capillary action depends on the cohesion between the liquid molecules and the adhesion between the molecules and the container. It is well-known that the inner shell of the microvascular has a uniform pore structure based on the previous measurement of the microvascular microstructure. The healing agent easily forms hydrogen bonds with the microvascular, determining that the repairing agent is more prone to capillary action after breaking the microvascular. It promotes the movement rate of the healing agent molecules, thereby improving the self-healing efficiency of asphalt [22].

### 3.2. Observation of Release and Diffusion Behaviours of the Self-Healing Agent

It is a critical premise that the healing agent in the biomimetic microvascular can only leak out in advance within reason before it works. The self-healing behaviour of asphalt depends on the release opportunity and release rate of the oily healing agents in microvasculars. Hydrogen bonds are formed between the PVDF microvascular molecule’s fluorine elements and the healing agent molecule’s hydrogen and oxygen elements. The formation of hydrogen bonds is beneficial for the oily repairing agent to fill the inner cavity of the microvascular evenly and release the healing agent. When the asphalt ages and triggers a microcrack extending toward the shell of the microvascular, the stress expands to puncture the microvascular. In an instant, the healing agent penetrates and is released under the action of hydrogen bonds through the inner micropores of the shell. Then, the healing agent moistens the asphalt’s small molecules. The molecular chains are connected. They tangle, restore the elasticity of, and then retard the ageing of the asphalt material.

Fluorescent indicators can indicate the diffusion rate and range for monitoring the diffusion and penetration of a liquid. This method is simple, non-invasive, reliable, and highly sensitive. In addition, this method can realise the real-time and dynamic observation of motion imaging and evaluation of liquid movement trends. The movement process of the healing agent could be observed under a fluorescence microscope in this work when the fluorescent indicator was mixed into the agent. The first issue to be considered was whether or not the sealing agent needs to be released continuously without stimulation. Figure 2a shows the state of the microvascular filled with a healing agent and fluorescent agent under the irradiation of an ultraviolet lamp. Figure 2b shows an image of a microvascular only containing fluorescent agents under ultraviolet light. Comparing the two images, it can be found that the microvascular containing the healing agent and the fluorescent agent is slightly darker than the microvascular containing only the fluorescent agent. Moreover, it can be determined that the healing agent and fluorescent agent do not leak out because no trace of diffusion of a fluorescent indicator out of microvasculars was observed. This scheme shows that it is feasible to rely on paraffin sealing. In addition, it was also proven that the wax seal at both ends of a microvascular could also avoid healing agent leakage.

Typically, self-healing agents can usually leak and be released in two ways. One method is to remove the healing agent through microvascular rupture. Another method is that the healing agent is slowly released through the micro-porous channels on the inner shell of the microvascular. The release modes of the agent can work together to improve the self-healing efficiency. This self-healing mechanism lays a foundation for future research on the relationship between the ratio of two different diffusion mechanisms and external forces. The self-healing process of the microvascular/asphalt composite system is the periodic continuous motion of the liquid self-healing agent, including its penetration, release, capillary, and diffusion steps. Diffusion is an important stage of the self-healing process, which causes the repair agent molecules to break, ostensibly extended to a greater extent. The self-healing action of microvascular/asphalt material can be regarded as a continuous movement process of a liquid healing agent. This movement process includes the release of the capillary, diffusion, and agent penetration. Among them, the diffusion process is a crucial stage of the self-healing process because it determines the movement rate of the healing agent molecules around the material cracks, which determines the self-healing rate of the material [22]. It is well-known that asphalt itself is a temperature-sensitive material. Therefore, the temperature is not only the main factor affecting the diffusion rate of the healing agent but also the main factor affecting the self-healing rate of asphalt material. Figure 3 shows SEM morphologies of the diffusion of a remediation agent into asphalt samples during self-healing at 0 °C. The cross-section morphology of the broken microvascular can be confirmed by observing the fractured surface of the asphalt. Figure 3a shows that the oily healing agent was contained inside of the microvascular and fractured microvascular, and the diffusion direction is marked with red arrows. Figure 3b shows that the diffusion zoom of the healing agent expanded by about 50 μm during 2 h. Based on the view of diffusion kinetics, the remediator molecules can diffuse based on concentration gradient kinetics, which indicates that the molecules tend to diffuse from high-concentration regions to low-concentration ones. The observation also proves that the different diffusion rates of the healing agent at other points leads to a wavy diffusion area, leading to a broader difbroadern range of the healing agent.

### 3.3. Effect of the Microvascular Contents on Self-Healing Efficiency

Microvasculars have always been used as reinforcement in composite materials, but the specific degree of support and toughening of asphalt materials has yet to be discovered. Significantly, the particular impact of microvascular-containing healing agents on the self-healing capability of asphalt mixture is also unclear. The SHE values of asphalt samples are explored using a designed tensile experiment. Figure 4a–d shows the cross-section images of asphalt samples with 4, 5, 6 and 7 microvasculars, respectively. The number of microvasculars can be counted from the cross-section morphology of samples. There was no interfacial separation between the microvasculars and the bitumen matrix. There was no mutual bonding effect between the various microvasculars. The fracture surface of the microvasculars appears circular without deformation. In all these images, it can be observed that the microvascular fracture surface is neat without adhesion. The above fracture characteristics of microvasculars facilitate the study of self-healing efficiency through the mechanical method.

Figure 5 shows microvascular asphalt samples’ original tensile load values at 0 °C. At the same time, both SHC_1_ (one-cycle) and SHC_2_ (two-cycle) values are calculated to compare the self-healing results. The first asphalt sample contains no microvascular, the second includes the initial hollow microvascular, and the third has the microvascular filled with a liquid self-healing agent. A single self-healing process of each sample was maintained at 24 h. By comparing the first tensile load of three pieces, it was found that the tensile load value of two pieces containing microvasculars exceeds far more than 110 N, and the tensile load value of asphalt samples without microvasculars also reaches 110 N, which is characteristic of the good toughness of asphalt material itself [22]. The tensile load values of the two samples with microvasculars are close and far exceed 110 N. Among them, the tensile load value of the sample containing hollow microvasculars is 115 N, and the tensile load value of the sample with a self-healing agent is 117 N. This can further prove the reinforcing effect of microvasculars in asphalt materials. In addition, it can be seen from the error bars that the SHC_1_ data deviation is less than the SHC_2_ data deviation. This indicates that the first self-healing process of microcracks in the material is more stable. This result is consistent with previous reports [8].

After the first tensile fracture, the three samples were put in a zero-temperature environment and completed a healing process for 24 h. After completing the first self-healing cycle, another tensile test was repeated. The fracture load values of the second tensile test (i.e., after the first self-healing process in the figure) were 56 N, 61 N, and 83 N, respectively. The SHC_1_ values of the three samples were 51%, 53%, and 71%. The self-healing capability of the sample with a healing agent was much greater than that of the other two without a healing agent. Therefore, it can be inferred that the healing agent’s movement is vital in the self-healing process. The sample was kept at 0 °C for 24 h after the second fracture, and the second self-healing process was carried out. Immediately, the above tensile fracture process was repeated. The tensile load values of the three samples were 51 N, 60 N, and 84 N. The SHC_2_ value was calculated according to formula (2). It was found that the SHC_2_ values of the two samples without healing agent decreased, and the SHC_2_ value of the samples without a microvascular had an even greater decrease of about 5%. Compared to that of the other two models (46% and 52%), the SHC_2_ value of the sample containing the healing agent remained above 70%, with an increase of about 1%. This may have been due to the complete diffusion and release of the healing agent in the microvasculars within 48 h. At the same time, the viscoelasticity of small bituminous molecules was restored under the softening action of the healing agent, the molecular chains were entangled again, and the self-healing capability was partially restored.

The SHC_2_ value of the microvascular-free asphalt was slightly less than its SHC_1_ value. The possible reason is that after repeated cold storage and fracture repair–re-fracture experiments, asphalt molecules are seriously aged, molecular entanglement is untied, intermolecular force is reduced, and performance becomes worse since its self-healing capability is damaged with a longer service time. The tensile load value and SHC value of the samples containing only a hollow microvascular were higher than those of pure asphalt samples due to strengthening and toughening effect of microvasculars. Comparing the tensile load of microvascular/asphalt composite samples with/without a healing agent shows that the healing agent can improve the self-healing capability of ageing asphalt samples. Moreover, the healing agent’s movement behaviour plays a vital role during self-healing.

There are many influencing factors in the process of producing biomimetic microvascular self-healing asphalt. Undoubtedly, the number of microvasculars must be a significant factor. Figure 6 shows the effect of the number of microvasculars filled with healing agents (1–3) on the tensile load values at 0 °C. In addition, SHC_1_ and SHC_2_ values of asphalt samples were calculated for the above two self-healing cycles. The microvascular distribution direction in asphalt is consistent with the tensile law of the asphalt sample. Other test conditions of the three samples are identical. The tensile load values of the original samples were about 108 N, 111 N, and 115 N. The difference in the tensile strength values of the asphalt materials mentioned above was caused by the difference in the number of microvasculars. It can be concluded that more microvessels lead to a more significant tensile strength value. In addition, when the three samples completed the first 24 h self-healing cycle, the tensile load values of the three samples were 58 N, 76 N, and 82 N, respectively. When the three samples completed the second self-healing cycle, their tensile load values were 44 N, 65 N, and 78 N. Based on the above matters, the SHC_1_ values of the three samples were 52%, 67%, and 73%. This shows that more microvasculars mean more healing agents with higher self-healing efficiency. However, biomimetic microvasculars cannot restore the tensile strength of asphalt to its initial state. This conclusion applies to applying microencapsulated healing agents in self-healing asphalt materials [22]. When the ageing degree is lower than the limit of the asphalt, the aged asphalt cannot recover its performance. In other words, its SHE value has a maximum threshold. The SHC_2_ values of the samples were 47%, 64%, and 70%, and the trend of the deal change was the same as that of SHC_1_. This further shows that the increase in microvasculars will improve the self-healing effect. It can also be found from the comparison data that the SHC_2_ value is significantly lower than the SHC_1_ one for the same asphalt sample. This shows that the self-healing efficiency of the material would be substantially reduced after a multi-cycle self-healing process. The reason for this result may be that the healing agent consumes a large part in the first self-healing process, and the remaining small part agent can be used in the second self-healing cycle. In addition, it can be seen from the error bars that the SHC_1_ data deviation was still less than the SHC_2_ data deviation.

### 3.4. Effect of Time on Self-Healing Efficiency

Time is one of the critical factors affecting the self-healing capability of bituminous materials [1,8]. Because viscoelastic molecules can adjust their state with a long enough time, bituminous materials need enough time to wet and diffuse molecules and entangle molecular chains [8]. In addition, the molecules of the healing agent also need a relatively long time to move to the bituminous material. A simplified experimental method was designed to study the influence of time factors on the self-healing process. In detail, several asphalt samples with identical structures had different self-healing times under specific conditions. Comparing the final self-healing efficiency values will reflect the impact of time. Figure 7 displays the tensile strength values of asphalt samples with three microvasculars at 0 °C during two self-healing cycles. SHC_1_ and SHC_2_ values are also listed and were calculated based on these mechanical data. The healing time of each self-healing cycle is 1, 3, and 5 days, respectively. Early research shows that a maximum of 5 days is long enough to complete a self-healing process [22].

The three asphalt samples had nearly the same initial tensile values of about 114 N. However, after the first self-healing cycle, their tensile load values sharply decreased to 30 N, 57 N, and 101 N. Correspondingly, their SHC_1_ values were 26%, 52%, and 88%. Furthermore, their SHC_2_ values were 22%, 48%, and 86% after the second self-healing cycle. Two conclusions can be obtained based on the analysis of the above data. The first conclusion is that the SHC values of asphalt samples decrease with the increase in the number of self-healing cycles because of the characteristics of gradual attenuation of asphalt. The second conclusion is that the self-healing effect of asphalt samples gradually increases with time during the same self-healing cycle. Other than the time factor, aged asphalt does not have self-healing capability without the external help of a healing agent [22]. The data analysis in this study shows that the asphalt sample can almost recover to more than 82% of its initial tensile strength after suffering two self-healing cycles over ten days. Prolonging the time can significantly improve the self-healing efficiency of the same asphalt sample under the same conditions. It is worth emphasising that even if the healing time is extended in each cycle, the SHC value of the same sample will not increase in the next self-healing cycle. The infinite extension of time cannot improve the SHC value endlessly. A further explanation is that each asphalt sample has a time threshold in each self-healing cycle. The factors that affect the specific value of the sample threshold are very complex, and this problem will be studied in the following work.

### 3.5. Effect of Temperature on Self-Healing Efficiency

As we all know, self-healing is generally defined as the automatic closing of cracks in materials without assistance [22]. Bitumen is a viscoelastic polymer material with temperature sensitivity. Naturally, bituminous molecules can recover microcracks through molecular movement and chain entanglement under certain conditions. This characteristic can be attributed to the unique viscoelasticity of bituminous molecules. The above conclusions can be drawn from analysing the fracture section morphology of asphalt materials after a tensile fracture. It has been proven that tensile loading forces an asphalt sample to deform until the sample breaks into two sections slowly. The two sections of the fracture are spliced together according to the shape of the fracture surface. Then, the two joined sections are put horizontally into an incubator with a specific temperature to complete a healing process. The healing process is undisturbed by airflow and external vibration. The two sections of the fracture sample will finally become a whole again through the movement and adhesion of molecules on the fracture surface.

Figure 8 shows the temperature-affected SHC values of asphalt samples with one, two and three microvasculars at 0 °C, 5 °C, 10 °C, 15 °C, 20 °C, and 25 °C. As asphalt is a temperature-sensitive material, it softens and become viscous at high temperatures, so the setting of too high a temperature is unsuitable for this tensile test. From the curve’s trend, at the same temperature, the more microvasculars there are, the more healing agents can be released, and the higher the self-healing efficiency is. This conclusion is entirely consistent with the previous test results. According to the data curve trend of the three samples, a higher temperature can increase the SHC value of each asphalt sample because higher energy can accelerate the molecular motion of the asphalt and repair agent and reduce the molecular motion resistance. The molecular movement of asphalt is intensified, the probability of molecular entanglement is increased, and the diffusion movement of the repair agent is also intensified, further improving the self-healing efficiency [22]. The asphalt samples with one, two and three microvasculars have SHC values of 26%, 53%, and 88% at 0 °C. It should be emphasised that multiple healing cycles reduce the self-healing ability of asphalt samples. The molecule reached a new equilibrium state after repeated self-healing at the same temperature. Without external help, the energy required for the self-healing process is much higher.

### 3.6. Preliminary Theoretical Analysis and Future Work

The above experiments proved that releasing and diffusing healing reagent molecules is the key to healing. This self-healing mechanism is illustrated in Figure 9. When an asphalt sample has microcracks or fractures, the microvasculars in it are damaged by the tip stress of the crack at the same time. The effluent healing agent rapidly diffuses into the surrounding bitumen, softening the fracture surface of bituminous molecules and then healing the cracks (Figure 9a). Figure 9b shows the process. With the extension of time, the repair agent continuously diffuses into the asphalt at a specific temperature (Figure 9c). Then, the healing rejuvenator diffuses continuously into the asphalt under a specific temperature with a time extension (Figure 9d). Based on the above analysis, a conclusion can be deduced that it is essential to design self-healing asphalt material to establish the mathematical relationship between the SHC value and the internal microstructures. This mathematical model also includes the influencing factors of the external environment, such as healing time and temperature. Moreover, this study reveals a preliminary mathematical relationship based on the structural parameters and the principle of self-healing, which is very important for the later experimental values and the data simulation of the healing process.

The diffusion rate of the healing agent is affected by temperature, viscosity, and the ageing degree of bitumen. In addition, the self-healing agent molecules diffuse into the asphalt, and the movement ability of asphalt molecules is also greatly enhanced [12,22]. Bituminous molecules seriously move through the penetration and entanglement areas between the chain segments making the cracks disappear gradually. It can be imagined that the relatively wide fracture gap needs more time to complete a healing process. At the same time, the mobility of bituminous molecules is also affected by the appearance of the healing agent. Increasing the amount of the healing agent will significantly increase the mobility of bituminous molecules, thus increasing the healing rate of microcracks.

Through a detailed analysis of the mechanism of self-healing of a microvascular/ asphalt composite, it can be seen that the SHC value of asphalt is affected by two factors, and its functional expression can be written as follows (3).
(3)$$SHC=f_{1}\left(m_{reju},T\right)$$

*m_reju_* is the amount of release out of the rejuvenator, and *T* is the ambient temperature.

In addition, another factor that should be considered when discussing the self-healing process and its rate is the diffusion behaviour of healing agent molecules in asphalt. The analysis of the self-healing mechanism proved that the asphalt self-healing process can be regarded as a continuous movement of the healing agent [12,22]. The constant process includes leakage, capillary action, penetration, and diffusion of the healing agent. The diffusion process is the primary step to determine the self-healing effect and rate [22]. In a previous study, Fick’s law was used to mathematically describe the diffusion behaviour of the healing agent in the asphalt matrix. Its diffusion coefficient can be defined as the ratio of the molar flux to the concentration gradient. Self-healing mechanism analysis shows that asphalt’s ageing degree and microstructure are the main factors affecting the diffusion coefficient. The diffusion coefficient function equation will be derived in future work by comprehensively considering various influencing factors. Its variables include the microstructure of microvasculars, the number of microvasculars, temperature, time, and the performance of the asphalt matrix. The proposed functional relationship will guide the structural design and practical application of asphalt self-healing materials.

## 4. Conclusions

In this work, various tensile tests were carried out to evaluate the self-healing capability of asphalt using rejuvenator microvasculars affected by microvascular content, self-healing time and temperature. Based on these analyses of this work, the following conclusions can be drawn:(1)Through the analysis of the self-healing mechanism of microvascular/asphalt composites, it is known that the microvascular content, self-healing time and temperature are three main factors influencing their self-healing deficiency.(2)Based on the analysis of ultraviolet light images and SEM morphology, the self-healing action of microvascular/asphalt material and the self-healing process can be regarded as a continuous movement process of a liquid healing agent.(3)There was no interfacial separation between the microvasculars and the bitumen matrix. The SHC_1_ values of the three samples were 51%, 53%, and 71%. The SHC_2_ value of the microvascular-free asphalt was slightly less than its SHC_1_ value. Comparing the tensile load of microvascular/asphalt composite samples with/without a healing agent showed that the healing agent can improve the self-healing capability of ageing asphalt samples.(4)With regard to the tension load values of microvasculars (three)/asphalt samples with a 1, 3, and 5 day healing time at 0 °C, their SHC_1_ values were 26%, 52%, and 88%. Furthermore, their SHC_2_ values were 22%, 48%, and 86% after the second self-healing cycle. Prolonging the time can significantly improve the self-healing efficiency of the same asphalt sample under the same conditions. It is worth emphasising that even if the healing time is extended in each cycle, the SHC value of the same sample will not increase in the next self-healing cycle.(5)The temperature-affected SHC values of asphalt samples with one, two and three microvasculars at 0 °C–25 °C imply that a higher temperature can increase the SHC value of each asphalt sample because higher energy can accelerate the molecular motion of the asphalt and repair agent and reduce the molecular motion resistance.(6)At last, this research reveals a preliminary mathematical relationship based on the structural parameters and principle of self-healing, which is very important for the obtention of later experimental values and the data simulation of the healing process in future work.

## Figures and Tables

**Figure 1 materials-16-04746-f001:**
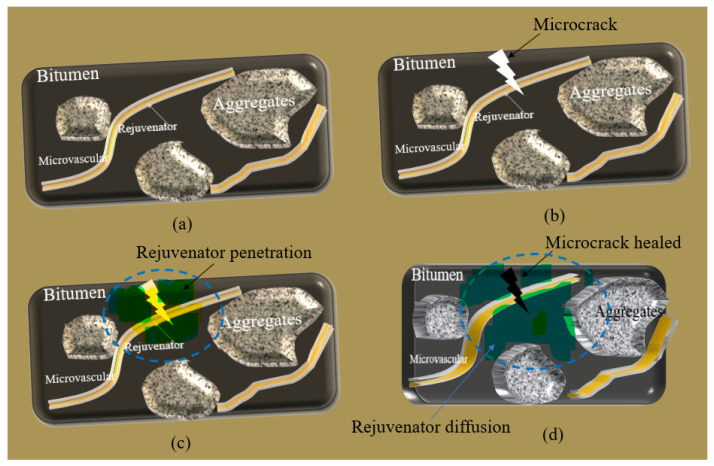
Illustration of process of microvasculars rejuvenating asphalt, (**a**) asphalt sample with multiple microvasculars, (**b**) microcrack triggering microvasculars and tip-stress of microcrack piercing microvasculars, (**c**) rejuvenator in microvasculars releasing and penetrating bitumen, and (**d**) microcrack healing and rejuvenator diffusion into bitumen.

**Figure 2 materials-16-04746-f002:**
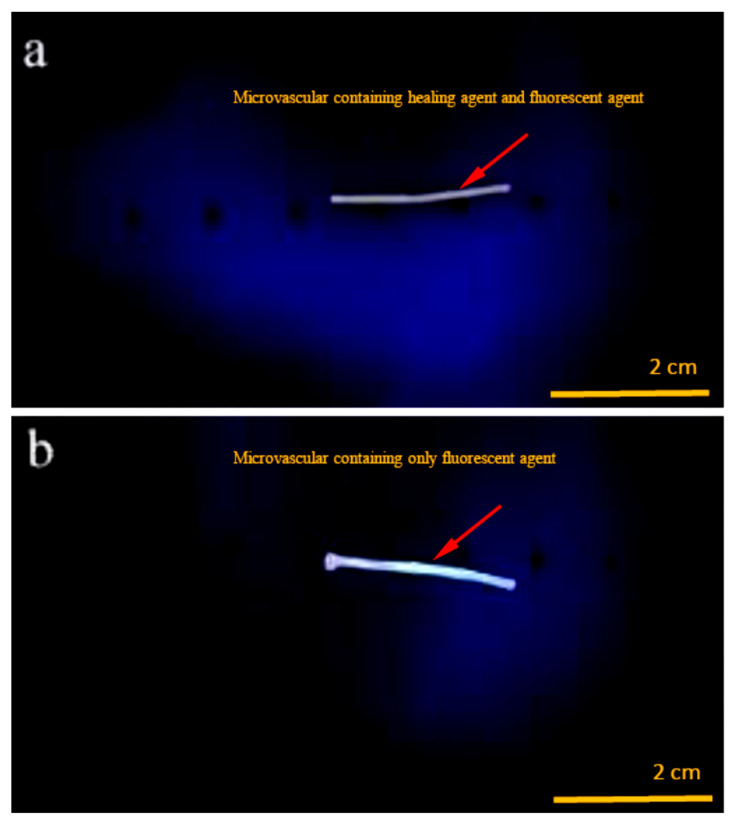
Images of a microvascular under ultraviolet light, (**a**) microvascular containing the healing agent and fluorescent agent, (**b**) and microvascular containing only the fluorescent agent.

**Figure 3 materials-16-04746-f003:**
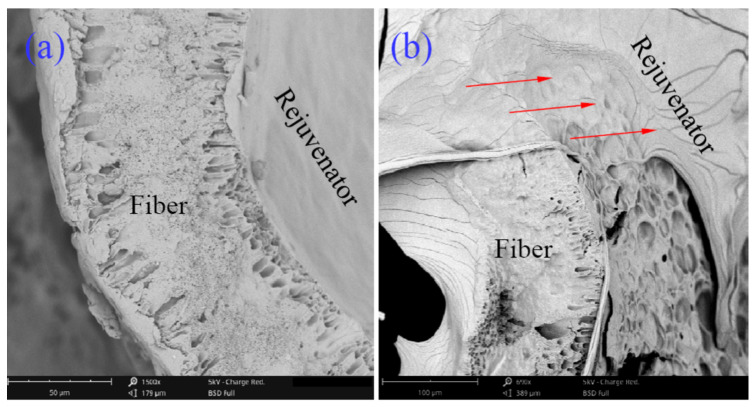
Fluorescence micrograph of rejuvenator diffusion into asphalt sample during a self-healing process, (**a**) oily rejuvenator diffusion into asphalt from a fracture microvascular, (**b**) oily rejuvenator diffusion rapidly into asphalt sample during 2 h and 4 h under 0 °C.

**Figure 4 materials-16-04746-f004:**
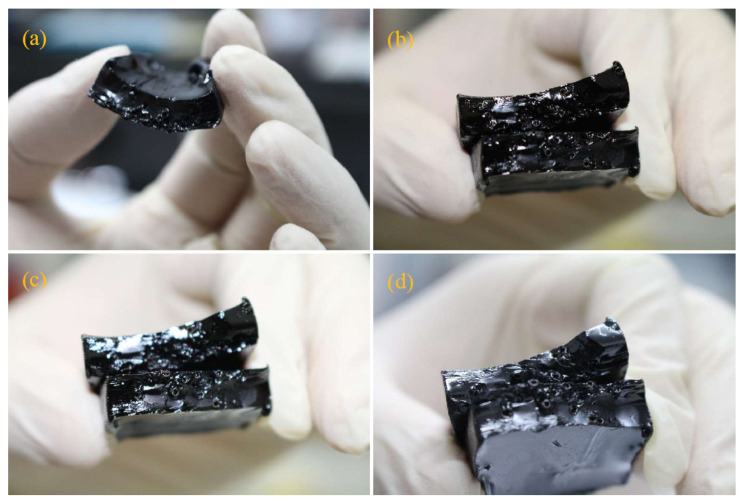
Cross-section photographs of asphalt samples with various microvasculars, (**a**) four microvasculars, (**b**) five microvasculars, (**c**) six microvasculars, and (**d**) seven microvasculars.

**Figure 5 materials-16-04746-f005:**
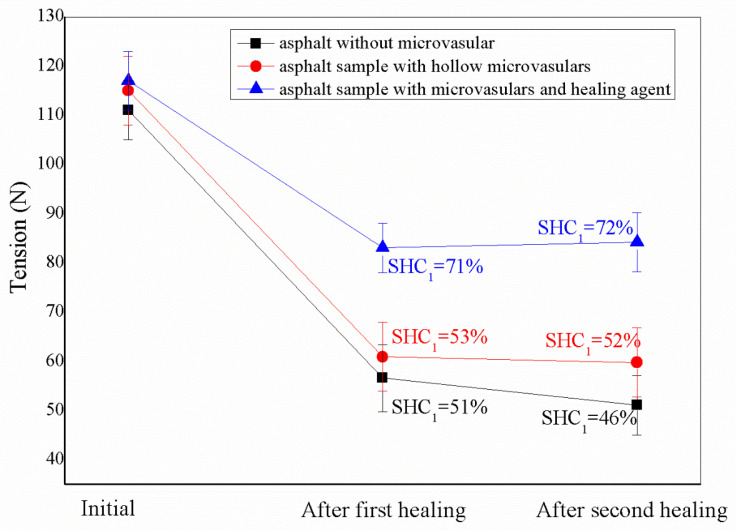
Tension values and self-healing asphalt samples include asphalt samples without microvasculars, asphalt samples with hollow microvasculars, and asphalt samples with microvasculars and healing agents; SHC_1_ and SHC_2_ values are listed.

**Figure 6 materials-16-04746-f006:**
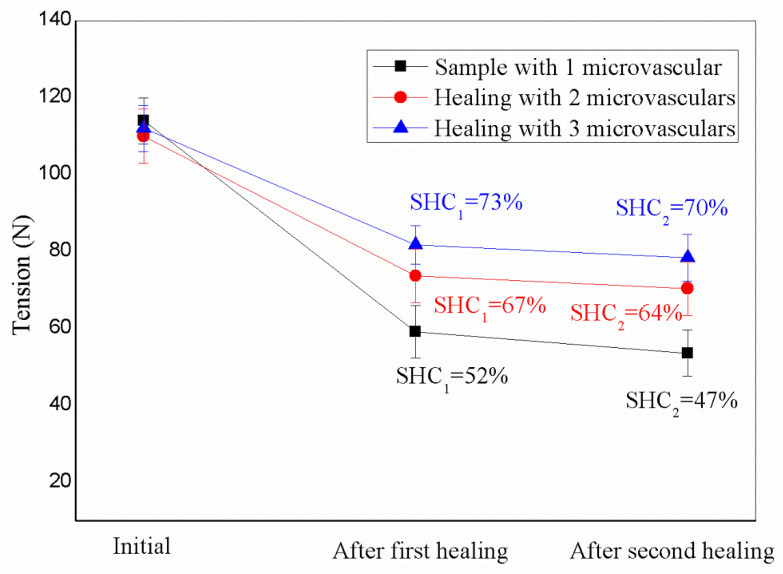
Tension load values of asphalt samples with 1–3 microvasculars.

**Figure 7 materials-16-04746-f007:**
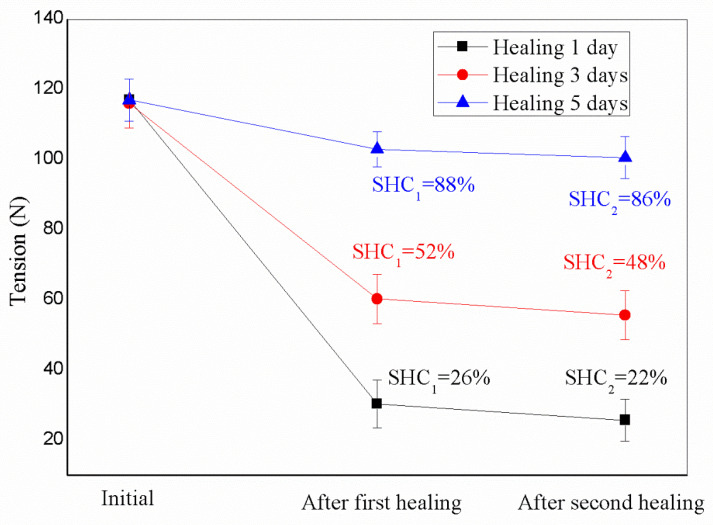
Tension load values of microvasculars (three)/asphalt samples with 1, 3, and 5 day healing time under 0 °C.

**Figure 8 materials-16-04746-f008:**
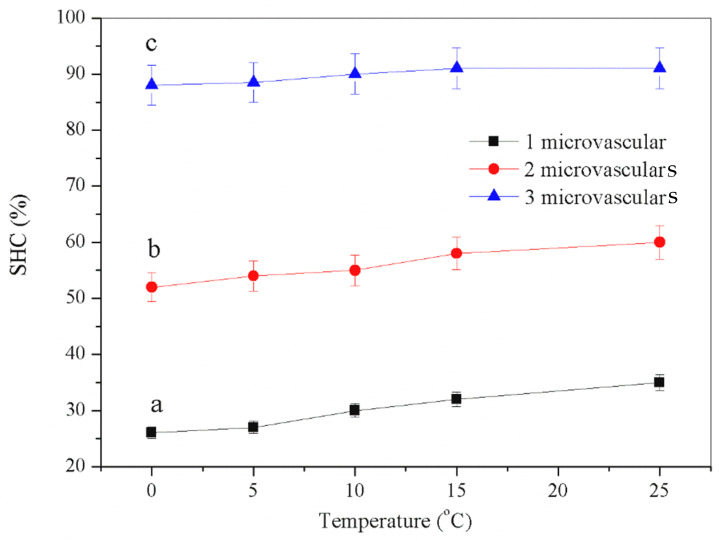
Self-healing efficiency (SHC) values of asphalt samples with 1–3 microvasculars influenced by self-healing temperature in one self-healing cycle.

**Figure 9 materials-16-04746-f009:**
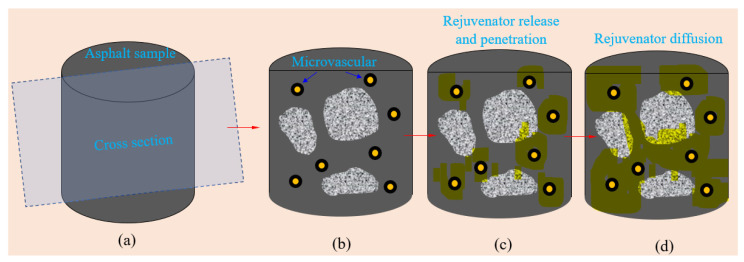
Illustration of the self-healing process of asphalt sample with microvasculars filled with a healing agent (rejuvenator), (**a**) cross-section of an asphalt sample, (**b**) rupture interface of an asphalt sample with broken microvasculars, (**c**) rejuvenator release due microvascular penetration into asphalt, and (**d**) continuous healing rejuvenator diffusion into the asphalt at a specific temperature with a time extension.

## Data Availability

Not applicable.

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
