# Peer review of "Investigation of Asphalt Self-Healing Capability Using Microvasculars Containing Rejuvenator: Effects of Microvascular Content, Self-Healing Time and Temperature"

_materials, 2023, doi:10.3390/ma16134746_

Round 1

Reviewer 1 Report

materials-2428987:

Follow my comments:

*Moderate review in English must be performed (by a native speaker).

*Title is confusing. Please clarify about "time and temperature".

*In the abstract the authors must present data (numerical values) to support the findings and the main conclusion of the paper.

*The term "theoretical analysis" is not a keyword because it occurs only a few times in the text.

*The introduction should make clear the contribution of the article to the field of knowledge. Contextualize more about the impact of this research on the frontier of knowledge.

In addition, important works on the subject are forgotten:

https://doi.org/10.1016/j.ijfatigue.2022.107440

*Lines 144-150: What is that? This information should not appear here.

*Lines 152-154: It is necessary to present the physical and chemical properties of the following materials: asphalt binder; PVDF; oil rejuvenator and dimethylacetamide.

Provide a complete characterization.

*2.2 Preparation of microvascular containing rejuvenator:

All parameters (time, temperature, pressure, number of turns, etc.) used in the preparation must be justified; the proportion of the compounds as well.

*2.3 Mixture of microvascular in bitumen:

This section should be explained in more detail. As it is, it's not enough.

*2.4 Observation of oily rejuvenator release and diffusion:

Has this analysis procedure already been used by other studies? Reference scientific publications.

*2.5 Design of test method for self-healing capability:

Authors must justify all parameters used in this procedure.

*3. Results:

-The findings must be confronted with the existing literature. Discussions are superficial.

-Indicate in figure 4 what is discussed in the text.

-The authors should discuss the results of figure 5 considering the standard deviation.

-Same as previous comment for the results in figures 6 and 7.

*5. Conclusions:

The authors replicate points from the discussions of the results in the conclusions.

The conclusion section must be concise and categorically respond to the general objective of the research.

Reduce the conclusions, be objective!

Moderate review in English must be performed (by a native speaker).

Reviewer 2 Report

Ref. No.: Manuscript ID: materials- 2428987

Title: Understanding of self-healing capability of asphalt using microvasculars containing rejuvenator: effects of microvascular content, time and temperature

This paper aimed to provide a practical and theoretical understanding of asphalt's self-healing capability using polyvinylidene fluoride microvasculars. Effects of effects of microvascular content, time and temperature are presented. I recommend publication once the minor comments below are addressed:

1) Improve quality of written English in Abstract.

2) In Introduction, the literature review could be improved by adding other important publications, such as:

·         Liang et al, 2021. Review on the self-healing of asphalt materials: Mechanism, affecting factors, assessments and improvements. Construction and Building Materials. Volume 266, Part A, 10 January 2021, 120453

·         Abadeen et al., 2022. Comprehensive Self-Healing Evaluation of Asphalt Concrete Containing Encapsulated Rejuvenator. Materials 2022, 15(10), 3672.

·         Wang et al., 2022. Self-Healing Properties of Asphalt Concrete with Calcium Alginate Capsules Containing Different Healing Agents. Materials 2022, 15(16), 5555.

3) The instructions on how the section should be written must be deleted from the text. See page 4, lines 144-149, and section 4 (Discussion).

4) Section 2.1 should be better presented.

5) It is better if the figure comes after the first reference in the text, especially when starting a section. Therefore, correct the positioning of Figure 1 by placing it after the first paragraph.

Minor editing of English language required. Especially in the abstract.

Reviewer 3 Report

Dear Authors,

Thank you for your work. Below, I'm sending some suggestions:

1. In the abstract, the authors should include at least the amount of the asphalt improver used.

2. Line 249: There are many factors affecting the mechanism of microvascular asphalt. - please list at least 3 factors so that this sentence is not so general, because the factors indicate a further research process / appropriate modification.

3.Line 491 - The authors indicate a temperature in the range of 0-25oC, at higher temperatures the bitumen becomes more 'sticky'. Research was carried out in this temperature range, but unfortunately in real conditions the temperature felt in city centers can exceed 40oC. As a result, the phase structure of the material changes and phases such as CSH crystallize to form more ordered structures (principles of thermodynamics), which may adversely affect the physical properties of the asphalt mass.

Are the authors able to write something more on this subject? What is the phase structure of the tested material and is it possible to predict how modified bitumen will behave at temperatures below 25oC?

It is also possible to indicate the scope or area of application of bitumen modified in this way (climate, region).

No discussion in the work - please complete this point, at least referring to the structure and microstructure of the material, because the self-healing modifier affects the internal structure first, and only then the physico-mechanical properties of the asphalt.

Please suggest potential simulations. Not every test can be done due to technical, technological or economic possibilities, but there are programs where, based on XRF (elemental composition) analysis, it is possible to simulate the behavior of the material under the influence of, for example, temperature (e.g. GEMS-PSI program).

The article is interesting in general, but the modification with a repair agent suggests more information on the interference and effect of the modifier on the internal structure of the asphalt.

Thank you and Regards,

Reviewer

Reviewer 4 Report

These are reviewers comments for the manuscript: Understanding of self-healing capability of asphalt composites using microvasculars containing rejuvenator: effects of microvascular content, time and temperature, submitted for publication in the MDPI Journal of Materials.

Authors did fine job I have enjoyed reading the material and finding out the progress in the extrinsic self healing system.  However I do have few small comments that I would like authors to consider:

1. Introduction , up to date extrinsic asphalt healing systems are not discussed, eg. the hybrid combination of induction and rejuvenation. Please refer to following publications:

i) Leegwater, G., et al, 2022, “Terms and Definitions on Crack-Healing and Restoration of Mechanical Properties in Bituminous Materials”. In: Di Benedetto, H., Baaj, H., Chailleux, E., Tebaldi, G., Sauzéat, C., Mangiafico, S. (eds) Proceedings of the RILEM International Symposium on Bituminous Materials. ISBM 2020. RILEM Bookseries, vol 27. Springer, Cham. https://doi.org/10.1007/978-3-030-46455-4_6

ii) Wan, P., et al., A novel microwave induced oil release pattern of calcium alginate/ nano-Fe3O4 composite capsules for asphalt self-healing. Journal of Cleaner Production, 2021. 297: p. 126721, https://doi.org/10.1016/j.jclepro.2021.126721

2. Remove guide to authors paragraphs:

i) section 2, pg 4. ln 144 - 150

ii) Section 4, pg 17, ln 560 -564

3. pg 6 ln 223: After the above mixture was cooled to room temperature - for how long the mix was cooled for, 1h, 2h, 4h, please specify.

4. Please give mix detail, mix composition. In Figure 4 samples are pure bitumen there are no aggregates in the mix. please clarify

5. Figures 5 - 8 are showing very high healing rate, are these pure bitumen samples or asphalt mix samples ( samples containing aggregates and bitumen)?

Reviewer 5 Report

Before making any recommendations for a very interesting scientific article “Understanding of self-healing capability of asphalt using microvasculars containing rejuvenator: effects of microvascular content, time and temperature”, I would like to present the following statements on the topic. Based on my long-term research in the field of holistic perception of asphalt pavements, I consider the evaluated scientific article to be topical and fully convergent with the following author's long term research and educational premise.  Asphalt pavements should be designed, built, managed, maintained, recycled (decomposed) at a reasonable price, in reasonable quality, respecting the relevant requirements of users, residents and sustainable development, including saving non-renewable resources and circular economy principles. In order to improve the contribution enabling its immediate publication in a renowned scientific journal Materials, I would like to recommend incorporation of the following mandatory and facultative recommendations.

Mandatory requirements:

LNSA (Line Number Scientific Article) 9-19….Abstract…  I would allow myself to take the following position regarding the abstract. In general,  I expect from a high-quality scientific abstract a brief scientific summary of the solved problem, an explicit determination of scientific goals and corresponding methodology. In the abstract  absent a specification of the application area (civil engineering, chemistry, pavement, ...)  in which asphalt's selfhealing is manifested and for which the evaluation results are presented. The sentence "The self-healing efficiency value indicated that increasing the temperature improved each sample's self-healing efficiency" needs to be restyled and indicated in which temperature range this statement applies. For the stated reasons, it is necessary to rework the abstract and clearly state which area of material and environmental engineering it refers to.

LNSA 25-26... However, under long-term ultraviolet, light, humidity, and other environmental conditions, the bituminous material's self-healing function appears to be weakened...in a scientific article clearly focused on a specific problem, it is not appropriate to use "and other environmental conditions". It would be appropriate to give specific examples or at least characterize the area of interest of the impacts.

LNSA 23-141…1. Introduction… this chapter needs to be significantly shortened, a possible way is indicated in the comment on the modifications of LNSA 560-564. Although it is not customary to present pictures in the Introduction, in the context of this article it would be worth considering moving some pictures, which would reduce the risk of psychosomatic fatigue for readers not working directly with asphalt self-healing.

LNSA 144-150… it is necessary to change the line spacing of the text, as it is in the entire article. Is the following text included in the article correctly? The Materials and Methods should be described with sufficient details to allow others to replicate and build on the published results. Please note that the publication of your manuscript implicates that you must make all materials, data, computer code, and protocols associated with the publication available to readers. Please disclose at the submission stage any restrictions on the availability of materials or information. New methods and protocols should be described in detail while well-established methods can be briefly described and appropriately cited.

LNSA 176-188…2.4 Observation of oily rejuvenator release and diffusion... I recommend considering the addition of illustrative photos facilitating the understanding of the text.

LNSA 190-191… It is found that the tensile fracture test is one of the effective methods to measure the self-healing efficiency of materials… it is necessary to state in the framework of which research the suitability of the mentioned test (the tensile fracture test) was determined and with which other tests it was compared.

LNSA 214…. (3) The test temperature is room temperature... it is necessary to justify whether the room temperature is sufficiently representative for the objectification of the interest ability of the interest rejuvenator.

LNSA 240-241… 3. Results, 3.1 Self-healing mechanism of microvascular/asphalt composites... Chapter titles need to be moved to the next page.

LNSA 560-564…4. Discussion... This 4-line chapter needs to be omitted, merged with the conclusion, or greatly expanded. In my opinion, the optimal solution with a slight text modification would be to move part of the introduction (LNSA 118-141) to this chapter, which would also fulfill the requirement to shorten the Introduction.

Facultative recommendations:

LNSA 2-4... Understanding of self-healing capability of asphalt using microvasculars containing rejuvenator: effects of microvascular content, time and temperature... I would like to recommend to the authors a slight modification of the title of reviewed scientific article. Personally, I would use, for example, the following title "Research of asphalt self-healing capability using rejuvenator microvasculars: effects of microvascular content, time and temperature"  (please take it only as an inspiration).

LNSA 155-169… 2.2 Preparation of microvascular containing rejuvenator … to this, for the wider professional public, it would be appropriate to add a diagram of the work progress or representative photos of the procedures to the text, which is difficult to understand.

LNSA 193-195… Microcrack is one of the leading causes of asphalt pavement damage. Understanding the behavior of crack healing is crucial to predicting the service life of asphalt pavement...I recommend the authors to list other causes of asphalt pavement damages too. A significant influence is also the climatic characteristics in the vicinity of the roads, which are significantly determined by the pavement's altitude. Details can be found, for example, in: Evaluation of the Effect of Average Annual Temperatures in Slovakia between 1971 and 2020 on Stresses in Rigid Pavements. Land, 11(6), 764.

LNSA 312-313… Figure 2. Images of microvascular under ultraviolet light, (a) microvascular containing healing 312 agent and fluorescent agent, (b) microvascular containing only fluorescent agent... it would be appropriate to increase the font size of the texts in the figure and   them more adapt  the visual of this  figure to the visuals another figures.

LNSA 424-469... 4. Summary and Conclusions ... I personally prefer to state the title of the chapter only in the form Conclusions or discussion and conclusions...A very good abstracting of the research findings has the potential for separate processing of chapters 4. Discussions and 5. Conclusions. In the conclusions, I recommend implementing a comparison of the author's research results with the most important works of foreign authors.

LNSA 543… T is temperature...it is necessary to add what temperature is involved (temperature of air, rejuvenator, asphalt mixture,...).

LNSA 570-572... At the same time, the existing perfect models are summarized, …in scientific articles it is not standard to write about perfect models (wives and husbands are perfect, but not models :)), were really: essential mathematical function relationship is analysed?

As an expert profiled in the field of holistic perception of civil engineering with an emphasis on circular economy and conservation of non-renewable resources in the field of asphalt pavements, I rate the article under review as excellent in the subject area. I would allow myself to repeatedly emphasize the requirement of shortening the introduction and supplementing schemes of research procedures and representative photographs facilitating the understanding of theoretically demanding texts. I congratulate the authors on the very good quality of the article, for us specialists in pavement engineering, it is an inspiring article. For the stated reasons, I will process the repeated review in a very short time, within 3 days at the most.

Round 2

Reviewer 1 Report

materials-2428987-R1:

Follow my comments:

*Minor editing of English language required (by a native speaker).

*Line 46-47: “Research has been carried out on the effect of mixed conductive additives on the distribution of steel fibers.” - Add the following reference: https://doi.org/10.1016/j.ijfatigue.2022.107440

*Review all the technical terms in the paper again, use terms from the test standards.

 Minor editing of English language required.

Reviewer 5 Report

Based on the incorporation of the changes recommended by me and credible justification their non-implementations ones,  I allow myself to rate the assessed second version of the scientific paper as follows. Reviewed contribution: "Investigation of asphalt self-healing capability using microvasculars containing rejuvenator: effects of microvascular content, (original title: Understanding of self-healing capability of asphalt using microvasculars containing rejuvenator: effects of microvascular content, time and temperature", I rate it as excellent. Based on my experience in the assessed issue and subsequent deepening of my knowledge, I am pleased that the submitted 2nd version of the article meets all my essential requirements for a quality scientific article. In conclusion, I would like to sincerely congratulate the authors on an excellent second version of reviewed scientific article and thank the publisher for the opportunity to expand my scientific knowledge in the following field. Tensile tests for  evaluate the self-healing capability of asphalt using rejuvenator microvasculars effecting by microvascular content, self-heading time and temperature. 
